# Acute Ischemic Stroke at High Altitudes in China: Early Onset and Severe Manifestations

**DOI:** 10.3390/cells10040809

**Published:** 2021-04-05

**Authors:** Moqi Liu, Mingzong Yan, Yong Guo, Zhankui Xie, Rui Li, Jialu Li, Changhong Ren, Xunming Ji, Xiuhai Guo

**Affiliations:** 1Department of Neurology, Xuanwu Hospital, Capital Medical University, Beijing 100053, China; liumoqi@ccmu.edu.cn (M.L.); lirui@mail.ccmu.edu.cn (R.L.); lijialu@ccmu.edu.cn (J.L.); 2Department of Neurology, Yantai Penglai Traditional Chinese Medicine Hospital, Yantai 622110, China; yanmingzong@126.com; 3Department of Neurology, Yushu People’s Hospital, Yushu 815000, China; cervical@sina.com; 4Department of Neurology, Huzhu People’s Hospital, Haidong 810500, China; xzk447888@126.com; 5Laboratory of Hypoxia, Xuanwu Hospital, Capital Medical University, Beijing 100053, China; rench@xwhosp.org; 6Department of Neurosurgery, Xuanwu Hospital, Capital Medical University, Beijing 100053, China; jixm@ccmu.edu.cn

**Keywords:** altitude, acute ischemic stroke, neuroimaging, risk factors, polycythemia

## Abstract

The detailed characteristics of strokes at high altitudes in diverse nations have not been extensively studied. We retrospectively enrolled 892 cases of first-ever acute ischemic strokes at altitudes of 20, 2550, and 4200 m in China (697 cases from Penglai, 122 cases from Huzhu, and 73 cases from Yushu). Clinical data and brain images were analyzed. Ischemic strokes at high altitudes were characterized by younger ages (69.14 ± 11.10 vs. 64.44 ± 11.50 vs. 64.45 ± 14.03, *p* < 0.001) and larger infract volumes (8436.37 ± 29,615.07 mm^3^ vs. 17,213.16 ± 47,044.74 mm^3^ vs. 42,459 ± 84,529.83 mm^3^, *p* < 0.001). The atherosclerotic factors at high altitude, including diabetes mellitus (28.8% vs. 17.2% vs. 9.6%, *p* < 0.001), coronary heart disease (14.3% vs. 1.6% vs. 4.1%, *p* < 0.001), and hyperlipidemia (20.2% vs. 17.2% vs. 8.2%, *p* = 0.031), were significantly fewer than those in plain areas. Polycythemia and hemoglobin levels (138.22 ± 18.04 g/L vs. 172.87 ± 31.57 g/L vs. 171.81 ± 29.55 g/L, *p* < 0.001), diastolic pressure (89.98 ± 12.99 mmHg vs. 93.07 ± 17.79 mmHg vs. 95.44 ± 17.86 mmHg, *p* = 0.016), the percentage of hyperhomocysteinemia (13.6% in Penglai vs. 41.8% in Huzhu, *p* < 0.001), and the percentage of smoking (33.1% in Penglai vs. 50.0% in Huzhu, *p* = 0.023) were significantly elevated at high altitudes. We concluded that ischemic stroke occurred earlier and more severely in the Chinese plateau. While the atherosclerotic factors were not prominent, the primary prevention of strokes at high altitudes should emphasize anticoagulation, reducing diastolic pressure, adopting a healthy diet, and smoking cessation.

## 1. Introduction

Cerebrovascular disease is one of major causes of death and disability around the world. In China, the death rate for cerebrovascular diseases was 149.49 per 100,000, accounting for 1.57 million deaths in 2018 [1,2]. Since China has a vast territory and diverse geographical environment, cerebrovascular diseases in different regions possess diverse characteristics. For instance, stroke incidence and prevalence generally follows a north-south and west-east gradient, with stroke rates almost three times higher in northern and northwestern China [3,4]. Although China has a large plateau area, cerebrovascular disease at high altitudes has not been extensively studied.

There is no consensus regarding the relationship between altitude and stroke because of the excessive confounding factors in the world, such as heterogeneous ethnicity, traditional cardiovascular risk factors, access to health care, migration, total altitude exposure over a lifetime, etc. [5]. Based on the above, strokes in each high-altitude area have their own characteristics due to differences in reginal climates, socioeconomics, and local traditional lifestyles. A large Swiss national cohort study conducted by Faeh and colleagues [6] and a study of U.S. hemodialysis patients at different altitudes conducted by Winkelmayer and colleagues [7] both suggested protective effects of living at high altitudes on stroke mortality and stroke prevalence. A possible mechanism for this may be related to hypoxia-inducible factor (HIF), which has conferred resistance against ischemia and improved function after myocardial infarction [8,9], as well as acting as a neuroprotective agent and a novel target for stroke therapy [10]. On the other hand, Jha and colleagues in Chadimandir, India [11], revealed that long-term residence at high altitudes was associated with a higher risk of stroke and large infarcts.

In China, about 290 million people live on the world’s largest plateau—the Qinghai-Tibet Plateau (366 km^2^), also known as “the roof of the world”, with an average altitude of above 4000 m. Therefore, altitude is a distinctive factor that cannot be ignored in the Chinese disease spectrum. An epidemiological survey conducted by the National Epidemiological Survey of Stroke in China (NESS-China) [4] showed that, compared with the average, the stroke prevalence, incidence, and mortality rates were higher in northwest China, where most highland areas are located. In addition, the life expectancy in Qinghai Province (where 84.1% of the area is above 3000 m) was 6.5 years below the national average. We hypothesized that cerebrovascular disease at high altitudes in China possesses special characteristics and a more severe manifestation compared with that seen in the plain. Our study was the first to describe and investigate the characteristics of acute ischemic stroke at different altitudes in China.

## 2. Materials and Methods

### 2.1. Geographical Background

This was a retrospective study. Clinical data were collected from three hospitals at different altitudes with latitudes between 30° and 40° in China. Penglai is a coastal city located in Shandong Province in eastern China. Huzhu and Yushu are both located on the Qinghai-Tibet Plateau in Qinghai Province, Central China. The geographic locations and features of the three districts are shown in Table 1 and Figure 1. These three regions are all county-level cities in China, with similar populations of around 400,000. The residents are mainly indigenous people, and the population is relatively stable. Penglai, Huzhu, and Yushu cover areas of 1007, 3400, and 13,462 km^2^, respectively, resulting in a lower population density at high-altitude areas. In addition, the ethnic composition of the three regions is different. Penglai is dominated by the Han nationality. Huzhu is a Tu Autonomous County, with a minority population—mainly Tu nationality—of 24%. Most people in Yushu are Tibetans, and other ethnic groups account for only 1.7%. Geographically, the altitude shows an average gradient of 20-2550-4200 m with similar latitudes.

We selected 3 major stroke centers from the above three locations: Yantai Penglai Traditional Chinese Medicine Hospital, Huzhu People’s Hospital, and Yushu People’s Hospital. They have similar levels of diagnosis and treatment and perform thrombolysis, mechanical thrombectomy, stent treatment, etc.

### 2.2. Study Population

All the patients admitted consecutively with first-ever acute ischemic stroke during the period 2018.1.1–2018.12.31 were included. First-ever acute ischemic stroke was defined as the acute neurological deficit for the first time within 7 days of onset, and imaging confirmed by Magnetic Resonance Imaging (MRI) or computed tomography (CT), with a hyperintense area on diffusion-weighted image (DWI) and a corresponding reduced signal on the apparent diffusion coefficient image (ADC) or a hypointense area on CT. Exclusion criteria included patients with neurological deficit caused by non-vascular causes, such as tumor, tuberculoma, trauma, etc.; patients with old lesions in the brain with a diameter of 1.5 cm or more; patients who had lived in the area for less than 2 years or had been outside the area for more than 2 months in recent 2 years. Detailed history was taken noting age, gender, and race, along with various risk factors. Biochemical investigations and imaging information (MRI and CT) were collected. The results of laboratory samples and vital signs were the first data after the patient was hospitalized. The imaging data were taken within one week of disease onset. We also noted the National Institute of Health stroke scale (NIHSS) at admission to hospital to comprehensively assess the severity of the clinical symptoms.

### 2.3. Imaging Processing

We processed MRI or CT images with ITK-SNAP [12] (version 3.8.0, Yushkevich et al.) to obtain infract volume data (Figure 2). Evaluation was conducted by two experienced neuroradiologists blinded to the patient’s clinical details. The average of the two values was noted.

We further differentiated ischemic infarction into lacunar infarct and non-lacunar infarct. Lacunar infarct was defined as rounded or ovoid lesions, >3 to <20 mm diameter, in the basal ganglia, internal capsule, centrum semiovale, thalamus, or pons [13,14,15]. Non-lacunar infarct was defined as stroke types other than lacunar infarct.

### 2.4. Risk Factors Analysis

The risk factors evaluated in this study include: hypertension (defined by a history of hypertension, or diagnosed at discharge), coronary heart disease (defined by a history of coronary heart disease, or diagnosed at discharge), diabetes mellitus (defined by a history of diabetes mellitus, or diagnosis at discharge, or glycosylated hemoglobin ≥7%), hyperlipidemia (defined by a history of hyperlipidemia, triglycerides ≥2.3 mmol/L, or total cholesterol ≥6.2 mmol/L) [16]. When further distinguishing hyperlipidemia, we defined hypercholesterolemia as total cholesterol ≥6.2 mmol/L and hypertriglyceridemia as triglycerides ≥2.3 mmol/L), hyperhomocysteinemia (defined by serum homocysteine >15 μmol/L), smoking (defined by a history of smoke), alcohol consumption (defined by a history of alcohol consumption), atrial fibrillation (defined by a history of atrial fibrillation, or diagnosed at discharge), and polycythemia (defined by hemoglobin ≥190 g/L in females or hemoglobin ≥210 g/L in males) [17].

### 2.5. Statistical Analysis

Categorical variables are presented as frequencies and percentages. Continuous variables are expressed as mean ± standard deviation. We examined the differences in categorical variables using χ^2^ or Fisher exact tests and the differences in continuous variables using a Kruskal–Wallis test or a one-way ANOVA. Analyses of covariance were further applied to adjust for age, gender, and race. In all the analyses, values of *p* < 0.05 were deemed statistically significant. Statistical analysis was performed with the SPSS software package (version 24.0, IBM Corp, Armonk, NY, USA). The results of the pairwise comparison are shown in Appendix A.

## 3. Results

### 3.1. Demographic Information

A total of 892 cases of ischemic stroke were included in this study: 697 cases from Penglai, 122 cases from Huzhu, and 73 cases from Yushu. The demographic information is shown in Table 2. The cases of ischemic cerebral infarction in the three regions were mainly male, and the male to female ratio was roughly 3:2. The average age of onset of ischemic stroke in Penglai was 69 years, while, in Huzhu and Yushu, it was 64 years, which signified that ischemic stroke occurred earlier in plateau areas than in plain areas. There were also significant differences in the ethnic distribution of cases in the three regions. Cases from Penglai were totally Han. The majority patients in Huzhu were Han (86%), and, in Yushu, they were Tibetan (92%).

### 3.2. Vital Signs

The vital signs of the patients with acute ischemic stroke at different altitudes are shown in Table 3. The respiratory rate, heart rate, and body temperature all increased with altitude, indicating a higher metabolism rate at plateau areas. The blood pressure will be discussed below.

### 3.3. Infarct Volume

A total of 676 MRI and 21 CT scans from Penglai, 93 MRI and 29 CT scans from Huzhu, and 51 MRI and 22 CT scans from Yushu were analyzed. The results are shown in Table 4. The histogram of the frequency distribution (Appendix B) shows that most patients suffered first-ever ischemic stroke with an infarct volume in the range of 0–50,000 mm^3^, which was 1/5 or less of the cortical volume of the hemispheres [18]. However, there were still a significant number of patients with catastrophic infarct volumes in the range of 50,000–500,000 mm^3^, which led to a skewed distribution of the infarct volume data in all three areas.

The infarct volume increased with altitude, and statistical significance was found between Yushu and other areas. After adjustment for age, gender, and race using the covariance method, the difference was still significant (*p* < 0.001). We further differentiated ischemic infarction into two groups, lacunar infarct and non-lacunar infarct, according to the imaging characteristics noted above. Lacunar infarct, usually caused by small vessel disease or perforating artery disease, accounted for 25.6% in Penglai, 36.9% in Huzhu, and 35.6% in Yushu (*p* = 0.014). In both groups, the infarct volume at high altitudes was significantly larger than that in the plain area, especially for non-lacunar infarcts.

### 3.4. NIHSS Score

The differences in the average NIHSS scores in three groups (3.90 ± 4.11 in Penglai, 4.70 ± 6.96 in Huzhu, and 5.15 ± 6.67 in Yushu) were not statistically significant (*p* = 0.483). We further analyzed the various components of the NIHSS score; the result is shown in Table 5. The incidence of a disturbance of consciousness was higher at high altitudes (6.31% in Penglai, 13.93% in Huzhu, 20.55% in Yushu), which, on the other hand, showed that ischemic stroke was more severe at high-altitude areas.

### 3.5. Risk Factors Analysis

The general distribution of risk factors for ischemic stroke patients at different altitudes is shown in Table 6. At high altitudes, diabetes, hyperlipidemia, and coronary heart disease were less prevalent in patients with ischemic stroke, while polycythemia, hyperhomocysteinemia, and smoking were more common. The prevalence of hypertension, alcohol consumption, and atrial fibrillation was similar. Note that, due to the lack of information in some of the medical records, data on the homocysteine levels, smoking, and drinking history in the Yushu area failed to be evaluated.

The prevalence of hypertension in patients with ischemic stroke at different altitudes was generally 69%. Most patients with hypertension were male, who accounted for 57.6% in Penglai, 64.1% in Huzhu, and 66.7% in Yushu (*p* = 0.375). Systolic pressure was lower in the plateau than in the plain (*p* < 0.001), while the result for diastolic pressure was the opposite, being higher in the plateau and lower in the plain (*p* = 0.016). There was no difference in mean arterial pressure among the altitude groups. Pulse pressure was lower at high altitudes (*p* < 0.001).

The prevalence of diabetes in ischemic stroke patients in the plateau was significantly lower than in the plain (*p* < 0.001). Patients at high altitudes had a lower level of glucose (*p* < 0.001) during the acute phase of ischemic stroke. A similar trend was also found in coronary heart disease, which was less prevalent in the plateau than in the plain (*p* < 0.001).

The prevalence of hyperlipidemia in patients with ischemic stroke at high altitudes was lower, especially in the Yushu area (*p* = 0.031). Patients with hyperlipidemia in Huzhu and Yushu mainly had hypertriglyceridemia (76.2%, 66.7%, respectively), while, in Penglai, they mainly had hypercholesterolemia (66.0%). The prevalence of hypercholesterolemia in the plateau was significantly lower than that in the plain. The difference was significant between Penglai and Yushu (*p* = 0.001).

In terms of blood lipids levels, differences were found between the levels of low-density lipoprotein cholesterol (LDL-C) and high-density lipoprotein cholesterol (HDL-C), and the total cholesterol level (CHOL). Patients in highland areas had a relatively lower level of CHOL (*p* < 0.001). The most favorable lipid level was found in patients in Huzhu, with the lowest LDL-C (2.45 ± 0.85 mmol/L) and the highest HDL-C (1.44 ± 0.38 mmol/L). Differences in the level of triglycerides were not found among the three groups (*p* = 0.242).

Polycythemia was a special risk factor for cardiovascular diseases at high altitudes. In our study, the prevalence of polycythemia in ischemic stroke patients in Penglai was 0, in Huzhu it was 13.9%, and in Yushu it was 9.6% (*p* < 0.001). The hemoglobin level, red blood cell count, and hematocrit were all significantly higher at high altitudes, indicating that polycythemia was an important issue in the plateau. The hemoglobin level did not show a significant difference between 2550 m and 4200 m above sea level.

The blood C reactive protein (CRP) levels were elevated in patients during the acute phase of ischemic stroke at three altitudes and were significantly higher in highland areas (*p* < 0.001). Differences were not found between 2550 m and 4200 m above sea level.

The serum homocysteine, smoking, and alcohol consumption data were only analyzed in Penglai and Huzhu because of the limited data for Yushu. The prevalence of hyperhomocysteinemia in ischemic stroke patients in these two areas was 13.6% and 41.8%, respectively (*p* < 0.001). In terms of the serum homocysteine level, we also found a higher level in high-altitude areas (*p* < 0.001). The number of smokers among ischemic stroke patients was significantly higher in Huzhu than in Penglai (50.0% vs. 33.1%, *p* = 0.023), while differences were not found among drinkers (27.3% vs. 28.1%, *p* = 0.093).

## 4. Discussion

Our study was the first to describe and investigate the characteristics of first-ever acute ischemic stroke at high altitudes. The age of onset of ischemic stroke in the plateau area was about 5 years younger than in the plain area in China, which suggested that the prevention and treatment of high-altitude cerebrovascular disease should be advanced.

The manifestation of the first-ever acute ischemic stroke in the plateau was more severe than that in the plain. Although the NIHSS scores did not differ significantly by altitude, disturbances of consciousness were more common in the highlands, with up to 1 in 4 patients presenting with impaired consciousness at the time of their first stroke at 4200 m. This manifestation was accompanied with a significantly larger infarct volume in the highland area. The significance was markedly detected in the non-lacunar infarct group, which indicated the more vulnerable involvement of large cerebral vessels. The characteristics of plateau lacunar stroke will be further analyzed elsewhere.

The risk factors for ischemic stroke at high altitudes were explained in detail. Except for heavy smoking and high homocysteine levels, the traditional risk factors for atherosclerosis did not play an important role at high altitudes in China. The prevalence of hyperlipidemia, diabetes, and coronary artery disease in patients with ischemic stroke was lower at high altitudes compared with in plain areas (*p* < 0.05). Although there was no difference in the prevalence of hypertension between the three areas, the systolic blood pressure was significantly lower at high altitudes (*p* < 0.001). The results of the fewer atherosclerotic risk factors found at high altitudes in our study were in accordance with previous reports [19,20,21,22]. It was also revealed that subjects from high-altitude areas had a lower plaque burden and less calcification in their carotid artery [23]. The reasons for this phenomenon involved people’s long-term adaptation to hypoxia, the stronger ultraviolet radiation at high altitudes, and there being less air pollution at high altitudes [23].

Furthermore, we found that at high altitudes, diastolic blood pressure conversely increased (*p* = 0.016), while the systolic blood pressure decreased, compared to that in plain areas (*p* < 0.001), resulting in a decreased pulse pressure at high altitudes (*p* < 0.001). This was mainly attributed to the rise in the heart rate and peripheral resistance caused by hypoxia-induced high metabolism and polycythemia [24]. On the other hand, the patients at high altitudes were younger in our study. Hypertension in young people is characterized by the activation of the sympathetic nerve system (SNS) [25] and renin–angiotensin system (RAS) [26,27]; meanwhile, the elasticity of the large arteries remains normal [28], thus leading to elevated diastolic blood pressure and systolic blood pressure remaining normal or slightly elevated [29,30,31]. The antihypertensive treatment strategy for high-altitude ischemic stroke should be more focused on reducing the diastolic blood pressure and preventing vascular remodeling.

Since the first-ever acute ischemic stroke at the plateau area in our study was characterized by a more severe manifestation and larger infarct volume with less atherosclerotic factors, we speculated that thrombosis caused by a hypercoagulable state might play a major role in ischemic strokes at high altitude. Polycythemia, also known as excessive erythrocytosis (EE), has been given extensive attention with respect to chronic mountain syndrome (CMS) [32]. The prevalence of polycythemia in Tibetans from Qinghai China is 1.21%, and in Han immigrants it is 5.6% [33]. In our study, the values in patients with ischemic stroke at high altitudes were around 10% (13.9% vs. 9.6%, *p* = 0.371). The hemoglobin level also significantly increased at high altitudes (*p* < 0.001). As a confirmed independent risk factor of cardiovascular disease [34], polycythemia reflects a maladaptation to the plateau environment, leading to an increase in peripheral resistance [35], a decrease in the velocity of blood flow [36,37], and a hypercoagulable state, thus promoting the formation of blood clots, mainly red thrombi, for which anticoagulant therapy is effective [38]. Therefore, as a perspective approach, the effectiveness and safety of anticoagulation treatments for patients with high-altitude ischemic stroke need to be further studied.

It was detected that the severity of stroke at high altitudes might also be associated with a strong inflammatory response in our study. Recent studies on pathophysiological mechanisms have increasingly confirmed acute ischemic stroke as a thrombo-inflammatory disease [39,40,41]. An elevated CRP level has been confirmed as an important risk factor for ischemic stroke [42], and is associated with greater severity and higher mortality [43]. In our study, the CRP levels were elevated in patients with ischemic stroke at three altitudes, and were particularly significant at highland areas, with a more than twice as high CRP level in Huzhu and Yushu. Hypoxia-induced inflammation [44] may cause stronger inflammatory responses at high altitudes, promoting thrombosis [45] and leading to severer strokes at high altitudes. The detailed mechanisms need to be further studied.

In addition, health education on the prevention of cerebrovascular diseases in plateau populations should be more focused on fruit and vegetable intake and smoking cessation. In high-altitude areas, the number of patients suffering from hyperhomocysteinemia (*p* < 0.001) and the average homocysteine levels increased significantly (*p* < 0.001). Due to the influence of the cold climate environment, hypoxia, and the influence of ethnic eating habits, people living in plateau areas consume relatively more meat and fewer vegetables. Therefore, the diet of high-altitude residents is high in lipids, high in cholesterol, and low in vitamins [46]. We suggested that people living at high altitudes consume more fruits and vegetables and reduce their fat intake for the prevention of cerebrovascular diseases. Smoking at high altitudes is another important issue worthy of attention. The number of smokers increased significantly in the plateau compared with that in the plain (50.0% in Penglai vs. 33.1% in Huzhu, *p* = 0.023). Therefore, people living at high altitudes should be advised to actively quit smoking.

There are limitations in our study. Firstly, although we distinguished lacunar infarct and non-lacunar infarct using imaging characteristics, the further evaluation of vascular imaging, such as computer tomography angiography (CTA) or magnetic resonance angiography (MRA), was limited due to economic constraints. Secondly, as a retrospective study, many social or geographical factors, such as occupation, sex, and religion, failed to be further analyzed. Thirdly, due to the low population density in highland areas, a relatively limited number of cases at high altitudes were enrolled. Rigorously designed prospective studies are further needed.

## 5. Conclusions

In China, ischemic stroke patients in plateau areas were characterized by younger age, more vulnerable to disturbances of consciousness, and larger infarct volumes than those in plain areas. While the atherosclerotic factors at high altitudes were not prominent, thrombosis due to hypercoagulative status might be an explanation, providing a theoretical basis for anticoagulant therapy for ischemic strokes at high altitudes. The primary prevention of ischemic stroke at high altitudes should emphasize reducing the diastolic pressure, consuming a healthy diet, and smoking cessation.

## Figures and Tables

**Figure 1 cells-10-00809-f001:**
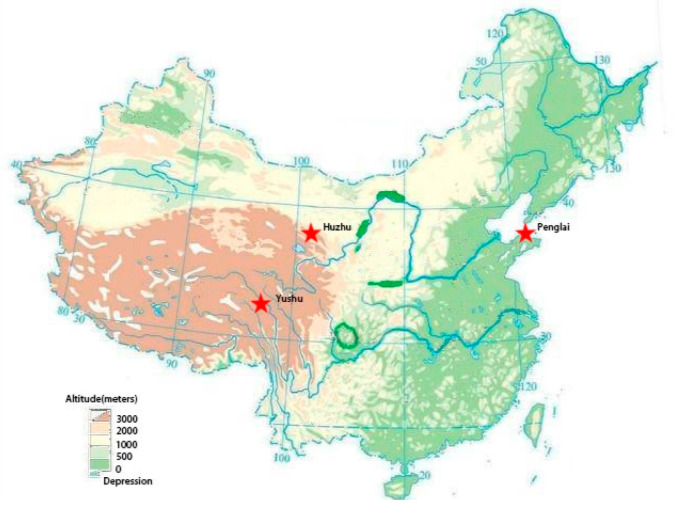
Topographic map of China and the geographical location of the three areas.

**Figure 2 cells-10-00809-f002:**
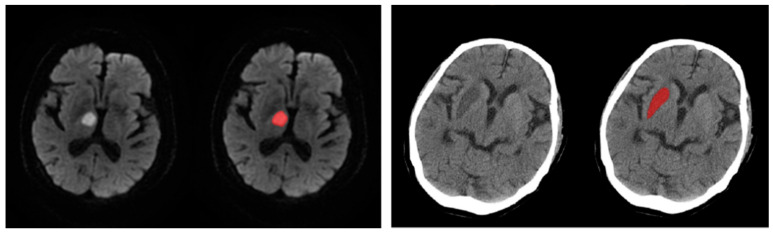
Using the software ITK-SNAP to obtain infarct volume by manual segmentation.

**Table 1 cells-10-00809-t001:** Geographical characteristics of the three areas.

	Longitude and Latitude	Annual Mean Temperature (°C)	Annual Mean Precipitation (mm)	Climate	Average Altitude (m)
Penglai	120° E, 37° N	11.9	606.2	Temperate maritime climate	20
Huzhu	102° E, 37° N	5.8	477.4	Temperate continental climate	2550
Yushu	97° E, 33° N	3	487	Plateau mountain climate	4200

**Table 2 cells-10-00809-t002:** Demographic characteristics of the three groups.

	Penglai	Huzhu	Yushu	*p* Value
Total	697	122	73	-
Gender				
Male (%)	410 (58.8)	81 (66.4)	49 (67.1)	0.140
Female (%)	287 (41.2)	41 (33.6)	24 (32.9)	
Age (years)	69.14 ± 11.10	64.44 ± 11.50	64.45 ± 14.03	<0.001
Race				
Han (%)	697 (100)	105 (86.1)	5 (6.8)	<0.001
Tibetan (%)	0	4 (3)	67 (91.8)	<0.001
Tu (%)	0	13 (11)	1 (1.4)	<0.001

**Table 3 cells-10-00809-t003:** Vital signs of the patients with acute ischemic stroke at different altitudes.

	Penglai (n = 697)	Huzhu (n = 122)	Yushu (n = 73)	*p* Value
Respiratory rate (times/minute)	18.49 ± 1.18	19.17 ± 2.62	20.71 ± 2.74	<0.001
Heart rate (beats/minute)	73.03 ± 10.79	77.53 ± 17.01	84.12 ± 17.37	<0.001
Body Temperature (°C)	36.37 ± 0.38	36.43 ± 0.31	36.58 ± 0.31	<0.001

**Table 4 cells-10-00809-t004:** Infarct volume of first-ever acute ischemic stroke at different altitudes.

Infarct Volume (mm^3^)	Penglai (n = 697)	Huzhu (n = 122)	Yushu (n = 73)	*p* Value
Total	8436.37 ± 29,615.07	17,213.16 ± 47,044.74	42,459 ± 84,529.83	<0.001
Lacunar stroke	728.09 ± 955.94	1896.18 ± 926.06	1957.52 ± 2367.42	<0.001
Non-lacunar stroke	11,116.73 ± 33,982.76	26,530.40 ± 57,720.96	65,377.57 ± 98,419.60	<0.001

**Table 5 cells-10-00809-t005:** Differences in National Institute of Health stroke scale (NIHSS) score components among the 3 areas.

KERRYPNX	Penglai (n = 697)	Huzhu (n = 122)	Yushu (n = 73)	*p* Value
NIHSS score	3.90 ± 4.11	4.70 ± 6.96	5.15 ± 6.67	0.483
Disturbance of consciousness	44 (6.31)	17 (13.93)	15 (20.55)	<0.001
Gaze	30 (4.30)	6 (4.92)	3 (4.11)	0.955
Vision	8 (1.14)	7 (5.74)	0.00	0.004
Facial paralysis	431 (61.84)	41 (33.61)	13 (17.81)	<0.001
Limb paralysis	410 (58.82)	63 (51.64)	37 (50.68)	0.169
Ataxia	80 (11.48)	13 (10.66)	1 (1.37)	0.027
Dysesthesia	115 (16.50)	30 (24.59)	12 (16.44)	0.093
Aphasia	56 (8.03)	14 (11.48)	11 (15.07)	0.085
Dysarthria	262 (37.59)	27 (22.13)	17 (23.29)	<0.001
Neglect	10 (1.43)	11 (9.02)	2 (2.74)	<0.001

**Table 6 cells-10-00809-t006:** Risk factors of patients with first-ever acute ischemic stroke at different altitudes.

	Penglai (n = 697)	Huzhu (n = 122)	Yushu (n = 73)	*p* Value
Hypertension [n (%)]	450 (64.6)	89 (73.0)	51 (69.9)	0.153
Systolic pressure (mmHg)	156.04 ± 23.35	142.75 ± 21.78	148.49 ± 25.99	<0.001 0.016
Diastolic pressure (mmHg)	89.98 ± 12.99	93.07 ± 17.79	95.44 ± 17.86	
Mean arterial pressure (mmHg)	112.00 ± 15.06	109.63 ± 18.06	113.12 ± 19.73	0.393
Pulse pressure (mmHg)	66.06 ± 17.44	49.69 ± 13.91	53.05 ± 14.75	<0.001
Diabetes mellitus [n (%)]	201 (28.8)	21 (17.2)	7 (9.6)	<0.001
Glucose level (mmol/L)	6.85 ± 3.01	6.09 ± 3.68	6.52 ± 2.59	<0.001
Coronary heart disease [n (%)]	100 (14.3)	2 (1.6)	3 (4.1)	<0.001
Hyperlipidemia [n (%)]	144 (20.2)	21 (17.2)	6 (8.2)	0.031
Hypertriglyceridemia [n (%)]	81 (11.6)	16 (13.1)	4 (5.5)	0.191
Hypercholesterolemia [n (%)]	95 (13.6)	8 (6.6)	2 (2.7)	0.018
LDL-C (mmol/L)	2.57 ± 0.77	2.45 ± 0.85	2.74 ± 0.91	0.028
HDL-C (mmol/L)	1.27 ± 0.32	1.44 ± 0.38	1.10 ± 0.35	<0.001
TG (mmol/L)	1.50 ± 1.40	1.44 ± 1.19	1.39 ± 0.80	0.242
CHOL (mmol/L)	4.94 ± 1.22	4.35 ± 1.13	4.27 ± 1.02	<0.001
Hyperhomocysteinemia [n (%)]	95 (13.6)	51 (41.8)	-	<0.001
Homocysteine level (μmol/L)	13.31 ± 8.76	15.82 ± 7.74	-	<0.001
Smoking [n (%)]	232 (33.1)	61 (50.0)	-	0.023
Alcohol consumption [n (%)]	196 (28.1)	12 (27.3)	-	0.093
Atrial fibrillation [n (%)]	57 (8.2)	11 (9.0)	3 (4.1)	0.462
Polycythemia [n (%)]	0	17 (13.9)	7 (9.6)	<0.001
Hemoglobin level (g/L)	138.22 ± 18.04	172.87 ± 31.57	171.81 ± 29.55	<0.001
Red blood cell count (*10^12^/L)	4.60 ± 0.59	5.40 ± 0.84	6.24 ± 1.00	<0.001
Hematocrit	41.59 ± 5.19	50.24 ± 9.42	49.02 ± 8.06	<0.001
CRP (mg/L)	7.92 ± 24.53	16.62 ± 16.40	18.67 ± 20.38	<0.001

LDL-C: low-density lipoprotein cholesterol; HDL-C: high-density lipoprotein cholesterol; TG: triglycerides; CHOL: Total cholesterol; CRP: C reactive protein.

## Data Availability

No new data were created or analyzed in this study. Data sharing is not applicable to this article.

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
