# Peer review of "Acute Ischemic Stroke at High Altitudes in China: Early Onset and Severe Manifestations"

_cells, 2021, doi:10.3390/cells10040809_

Round 1
Reviewer 1 Report
The authors analyzed 892 cases of patients living respectively at 20 (Penglai; n=697), 2550 (Huzhu; n=122) and 4200 meters (Yushu; n=73) above sea level. The main findings are that ischemic stroke patients at plateau areas (Huzhu, Yushu) were characterized by younger age, more severe clinical manifestations and larger infarcts (lacunar and non-lacunar) in relation to those at the plain areas (Penglai). The analysis of some risk factors like diabetes, hyperlipidemia and coronary heart disease were less prevalent in the patients at high altitude; however, polycythemia and hyperhomocysteinemia showed higher values in theses patients. In addition, the prevalence of hypertension, alcohol consumption and atrial fibrillation were similar among the three groups.
The study yield valuable information in relation to ischemic stroke and altitude, although it have some important limitations, some of them, even indicated by the own authors, but others not explicitly discussed. This is the case of the different number of patients evaluated at each altitude as well as the lack of some important data in the small population evaluated at Yushu area. However, the conclusion about the hypercoagulative status of patients at high altitude, as the main reason to explain the worse prognostic of these patients and the opening possibility to consider an anticoagulant therapy for them, are both interesting contributions.
It is striking that the volume of the infarcts (expressed as mean ± SD) show mean values much smaller than its corresponding standard deviations (table 4); therefore, the authors must include an explanation of these data. Finally, a careful language check should be done.
Reviewer 2 Report
Chinese scientists conducted a very interesting study to assess the incidence of ischemic strokes depending on the height above sea level. The authors observed a significant effect of altitude on the demographic and clinical characteristics of patients with ischemic stroke.The manuscript is well written, but requires major revision.
I have attached notes to the manuscript below:
1. Introduction:
a. it is not necessary to provide the first names of the authors of the articles cited.
b. authors should cite original studies on the relationship between HIF and the cardiovascular outcome.
c. the abbreviation NESS-China should be clarified.
d. other authors observed that high altitude prevents the occurrence of a stroke, while the authors of the manuscript hypothesize that it may cause a severe stroke - why this hypothesis? Please explain.
2. Methods:
a. L82-83 - Authors should be more specific.
3. Results:
a. the number of patients varies significantly between the subgroups. The authors should discuss this fact in the light of the results obtained.
b. authors should identify when NIHSS was measured - at admission to hospital?
c. table 4 should be redrafted - it is currently illegible.
d. table 7 should be combined with table 6.
4. Discussion: It is well known that ischemic stroke is a thromboinflammatory disease (doi: 10.1177/0004563216663775; doi: 10.3390/medicina55070342). The authors should discuss the possible association of altitude with hemostasis, inflammation, and the pathogenesis of ischemic stroke.
Reviewer 3 Report
The present study investigated the characteristics of ischemic stroke in three regions with different altitudes in China. It showed that patients with stroke in plateau regions have larger strokes and are younger than those in plain regions. The study is very well organized and focused on stroke features in less investigated regions. However, there are some points that should be considered:
- The groups have a wide difference in sample size; so, are the data normally distributed?
- In the conclusion section, it was mentioned that stroke patients at plateau regions have severe clinical manifestation but the results showed no significant differences in NIHSS score among the regions of different altitudes. Therefore, the conclusion section should be corrected.
- Is there any previous study that showed the epidemiology of stroke in these three regions? Giving that the population in these regions is similar, can we suppose that prevalence of stroke in Penglai higher than that in Huzhu and Yushu?
